# Effects of a Combined Method of Normobaric Hypoxia on the Repeated Sprint Ability Performance of a Nine-Time World Champion Triathlete: A Case Report

**DOI:** 10.3390/bs14111084

**Published:** 2024-11-12

**Authors:** Adrian Gonzalez-Custodio, Carmen Crespo, Rafael Timón, Guillermo Olcina

**Affiliations:** Faculty of Sport Science, Universidad de Extremadura, Av. Universidad, s/n, 10003 Cáceres, Spainrtimon@unex.es (R.T.); golcina@unex.es (G.O.)

**Keywords:** muscle oxygen saturation, hypoxia, repeated sprint ability, triathlon, case report

## Abstract

Elite athletes are an under-represented population in scientific studies, and there are no works analysing the influence of hypoxia in elite triathletes. The aim of this study was to analyse the influence of different methods of normobaric hypoxia on repeated sprint ability (RSA) performance. This study was a case study with an elite triathlete who has won nine triathlon world championships. The study used a combination of different methods of normobaric hypoxia. The three methods combined were as follows: live high-train low interspersed; intermittent hypoxic training; and intermittent hypoxic exposure. This study analysed the influence of these methods on RSA performance in variables such as power output, saturation of muscular oxygen, heart rate and ventilatory variables (VO_2_ and VCO_2_). The triathlete was measured before the training protocol (PRE), just after (POST-D3) and 21 days after the end of the protocol (POST-D21). This type of protocol has shown that it can lead to an improvement in RSA performance in the number of sprints (PRE vs. POST-D3 vs. POST-D21: 19 vs. 24 vs. 28), power output (PRE 615 W vs. POST-D3 685 W vs. POST-D21 683W) and efficiency of the triathlete. This work may be useful in improving power output and repeated sprint ability for elite triathletes.

## 1. Introduction

Triathlon is an endurance sport in which triathletes perform a total of three disciplines (swimming, cycling and running) sequentially. Elite athletes are considered to be those ranked among the top 125 athletes in the International Triathlon Union (ITU) world ranking [1]. Some scientific studies have analysed elite triathletes, but there is insufficient scientific literature on top-class elite triathletes. It is extremely difficult to work with this population’s characteristics because they have to maintain all their routines. There are different distances or challenges that can be undertaken in a triathlon: Ironman, Middle Distance, Olympic Distance and Sprint. It is important to note that in the Olympic and Sprint distances, bike drafting is allowed by the ITU. During a draft-legal triathlon, there are some important variables that influence performance. The contemporary triathlon, particularly in the ITU events, has some common characteristics over the bike course. There are some difficulties related to breaking up big groups and making the cycling tougher than on a flat course. These difficulties are usually related to technical issues, which make the bike perform intermittently with some supra-VO^2^ Max or sub-VO^2^ Max effort [1]. The difficulties over the bike course require the athlete to have the ability to produce intense short efforts separated by short recovery periods, which has been termed “repeated sprint ability” (RSA) [2]. These physiological requirements make the elite athlete train using intermittent efforts. Elite triathletes sometimes train using the hypoxia method—usually live high-train high (LHTH) or live high-train low (LHTL) [3]. Other studies show that combining methods is the best way to improve performance [4]. One of the methods that has been shown in the literature to enhance performance is repeated sprint training (RSA). RSA performed in ambient hypoxic conditions (RSH) produces greater improvements than training at sea level [5]. The aim of this study was to analyse the influence of different methods of normobaric hypoxia on RSA performance. The main goal of this protocol is to improve the performance of RSA.

## 2. Materials and Methods

### 2.1. Participant

This is a case report in accordance with the CARE guidelines, included in the Appendix A, [6] presented with an elite triathlete. The participant had won nine World Championships (2× Middle Distance, 5× ITU Distance and 1× Xterra Triathlon) and an Olympic silver medal. The main objective of the triathlete in this study is the ITU season, with the World Series being the most important event in the year. The participant was 34 years old, 178 cm tall and weighed 73.3 kg, with a body fat percentage of 8.05% and a total of six site skinfold measurements of 45.5 mm. The athlete was measured before the protocol in order to adjust the load and the training stimulus accordingly. For the measurement, an incremental ergometric test was performed on a bike until exhaustion [7].

### 2.2. Study Design

The study started at the beginning of the triathlon season. During the protocol, the triathlete developed an average training regime of 25,000 m of swimming, 400 km of cycling and 90 km of running; this training load was still the same 21 days after the protocol, between the hypoxic stimulus and the last test. The triathlete had performed a combined protocol of normobaric hypoxia over a total of four weeks. The first method was live high-train low interspersed (LHTLi). The participant slept for five days in a row with SpO_2_ increased progressively to up to between 94% and 90% of SpO_2_. Then, he rested for two days of sleep in hypoxic conditions. The second protocol was intermittent hypoxic exposure (IHE), which consisted of giving a stimulus of 14.40 to 13.20 SpO_2_ every day, decreasing progressively during the protocol to leave the SpO_2_ of the triathlete between 88% and 90%. The last protocol was intermittent hypoxic training (IHT) with a total of eight sessions, which comprised two to three sets of five sprints all out for 10” with a rest of 20” between sprints and between sets of 5’ (the first three sessions consisted of two blocks, followed by three sessions of three blocks and finally two sessions of two blocks). Between sessions, the triathlete rested for 72 h of RSH. The SpO_2_ during the training was between 13.5% and 14.0%. The protocol was performed fully and without any changes without any adverse or unanticipated events (Table 1).

### 2.3. Assessment

The protocol used to measure the increase in performance was an RSA protocol, which was performed three times: one before the intervention started (PRE), the second 3 days after the protocol had finished (POST-D3) and the last one 21 days after the workouts had finished (POST-D21), because there is another improvement in performance 21 days after a hypoxic protocol [4].

The test consisted of doing the maximum number of 10” sprints, resting for 20” between each one until the triathlete reached voluntary exhaustion—we were working with an elite triathlete who knows exactly when total exhaustion has been reached. The triathlete warmed up before the test for 10’ at 65% of functional threshold power (FTP) and then performed five all-out sprints of 30”, resting for 3’ between them. When the triathlete had finished that block, he rested for 5 min and then started the test.

### 2.4. Variables

Skinfolds were measured by an ISAK-certified investigator with a plicometer (SATA, AMEFDA, Sevilla, Spain). Six skinfold measurements were taken and expressed in millimetres (abdomen, triceps, subscapular, supra-iliac, quadriceps and medial calf); weight was measured in kilograms with a precision of 0.1 kg (SECA 769, GmbH & Co. KG, Hamburg, Germany) and height was measured with a stadiometer in centimetres (SECA 769, GmbH & Co. KG, Hamburg, Germany). These variables were measured to characterize the participant.

The variable measured during the test was the volume of oxygen consumption (VO_2_) (L/min), measured with a Metalyzer 3B+ (Metalyzer 3B, Cortex, Germany) following the manufacturer’s instructions for calibration, and the data were analysed by MetaSoft Studio (MetaSoft Studio, Cortex, Germany). Heart rate (HR) (beats per minute) was measured with an H7 Belt (Polar, Finland) connected by Bluetooth to MetaSoft Studio 5.16.0.

Power was measured with a Quarq D-Zero (Sram, Chicago, IL, USA, EEUU) power meter installed on the subject’s own bicycle. We used average (AvgW) and maximum (MaxW) power in watts (W).

The subject was measured by near-infrared spectroscopy (NIRS) technology using a MOXY Monitor (MOXY, Hutchinson, MN, USA, EEUU) in the right vastus lateralis following the protocol of [8]. The variable that we used was muscle oxygen saturation (SmO_2_), which was shown in percentage (%). The SmO_2_ was analysed by comparing the highest (reoxygenation) vs. the lowest saturation after the sprint (ΔS_m_O_2_).

There were some variables regarding the number of sprints and for analysing how fatigue affects the performance of the participant. The first variable to be analysed was the number of sprints that the triathlete performed during the test. The fatigue index (FI) was calculated as a comparison of the best and worst sprint performance during the RSA test [9].
$$FI=\frac{\left(S_{Best}-S_{Worst}\right)}{S_{Best}}\times 100$$

Another index was used to analyse the performance in the RSA test—this was the percentage decrement score (*S_Dec_*). The *S_Dec_* compares the performance with an ideal performance where every sprint has the best power output.
$$S_{Dec}=\left\{1-\frac{(S_{1}+S_{2}+S_{3}+\ldots +S_{Fin}}{S_{Best}\times number\;of\;sprints}\right\}\times 100$$

## 3. Results

The results of the variables that analyse the performance in the RSA test are presented in Table 2. The data are based on different variables in analysing fatigue and performance during an RSA test.

The power output performance was analysed using the average power output during every sprint of the test (WAvg) (Figure 1). Heart rate (HR) (Figure 2), volume of oxygen (VO_2_) (Figure 3) and the difference between maximum and minimum muscle oxygen saturation (ΔSmO_2_) (Figure 4) was compared by the test before the training protocol (PRE), 3 days after the protocol (POST-D3) and 21 days after the altitude training had finished (POST-D21).

The power output average was greater 3 days and 21 days after the protocol. The difference between 3 and 21 days was nearly the same, with more differences between sprints POST-D21 and more sprints performed POST-D21.

The heart rate was higher just after the protocol (POST-D3), especially in the final part of the test, but there was not much difference between PRE and POST-D3; the difference was greater between POST-D3 and PRE vs. POST-D21. The heart rate 21 days after the protocol had finished was lower than it was PRE and POST-D3.

The oxygen consumption (VO_2_) had a similar tendency to the heart rate. PRE and POST-D3 were similar, but POST-D21 had a somewhat lower oxygen consumption.

The difference between SmO_2_ is greater at POST-D21 than POST-D3, and both these were greater than PRE. The difference between POST-D3 and PRE was smaller when the RSA test was finished, and greater at POST-D21.

## 4. Discussion

The present study has shown suitable benefits of combined methods in normobaric hypoxia: live high-train low interspersed (LHTLi), intermittent hypoxic exposure (IHE) and repeated sprint ability in hypoxia (RSH). One of the main findings that we have seen in this study is the increase in the number of sprints that an elite triathlete can achieve until exhaustion. The results show that the number of sprints increased just after the training protocol and 21 days after this protocol came to an end (19 vs. 24 vs. 28), which means that RSH increased the capacity of an elite triathlete to replicate maximum efforts over a longer time. In relation to this increase in the ability to generate changes in rhythm and repeated sprints, one of the factors that affects the success of a short-distance triathlete is the capacity to perform intermittent and very intense efforts required due to the characteristics of the bike course, as well as the current needs of the competition, in the triathlon cycling sector [1]. Ensuring the response and adaptations in this 21-day training programme also allows the progression of the training load to be controlled in the protocols for altitude training camps with triathletes of this level of performance [4].

Some studies show that the number of sprints that an athlete performs depends on the kinetics of the oxygen consumption of the athlete [10]. The triathlete measured had really fast VO_2_ kinetics, which was improved after the hypoxia protocol, which is one of the arguments that makes it possible to do more sprints between PRE, POST-D3 and POST-D21 [5,11,12]. In other studies, the number of sprints depends on how the muscular saturation of oxygen changes during the effort expended and rest (SmO_2_) [13]. In this study, the average SmO_2_ increased at POST-D3 and POST-D21 (12.63% vs. 15.04% vs. 20.17%), the same as the number of sprints; so, an elite triathlete with a greater SmO_2_ will be able to develop intermittent sprints more often.

The average power output in the sprints increased after the training protocol, which means that the repeated sprint training improved performance in the elite triathlete. The power output decreased slightly 21 days after the protocol had finished (685 W vs. 683 W) but the triathlete performed four more sprints, and after sprint number 18, the power output was bigger at POST-D21 than at POST-D3 training (POST-D3 = 666 W vs. POST-D21 = 686 W); so, this study confirms that the hypoxic stimulus increases performance in the fatigue condition. The literature has several hypotheses to explain this. The first one depends on the performance of the first sprints (PRE = 627 W vs. POST-D3 = 687 W vs. POST-D21 = 658). The power output in the first sprints correlates positively with a decrement in performance over the last sprints [9]. It could be interesting for futures studies with elite triathletes to measure the electromyography in order to obtain data on neuromuscular fatigue. Combining near-infrared spectroscopy (NIRS) and surface electromyography (EMG) will provide complementary information to help in understanding the reasons for the athlete becoming exhausted [14].

VO^2^ and HR are lower at POST-D21 after training than in the PRE and POST-D3 training protocol, although the ΔSmO^2^ is higher. The increase in this variable is presumed to be due to a greater perfusion of the athlete’s muscle fibres because there is not a better consumption of oxygen (VO^2^). Other studies confirm the argument that RSH does not enhance the aerobic performance of an elite endurance athlete [15], but some studies show better oxygen perfusion in the different muscular fibres after a hypoxic stimulus [16]. In addition, there are some studies that show benefits in the muscle buffering, enhancing performance in short and repetitive efforts [17]. Future studies could include some analysis of how the muscle fibre changes after this protocol, but it is difficult to apply that invasive technique to elite athletes.

One limitation of this work is that it does not measure a control group, so it is difficult to confirm that all the changes are caused by the hypoxia stimulus. There are no other studies with this subject characteristic or other athletes who could be compared. Other studies show bigger improvements in VO^2^ Max during repeated sprinting in hypoxia than in normoxia [5,18]. VO^2^ Max is the main performance variable in technical cycling courses in triathlons [1]. Another important point is that the athlete’s normal training could interfere with the hypoxic protocol, but elite athletes need to continue their training to achieve their objectives, and during their season they will not only follow the RSH protocol. The interference of the normal training should be controlled, taking into account the training load.

This work proposes a protocol of combined methods in normobaric hypoxia for an elite triathlete to improve performance. It could be a useful protocol to use a new stimulus for the competition. The use of an RSH protocol will be more important when the upcoming competition includes a bike course with a lot of changes in intensity (technical courses). Future studies should measure more subjects against an organized control group to confirm the effects of the hypoxic stimulus. Another pathway that future studies could analyse in depth is the influence of the genetic code regarding how hypoxia affects elite triathletes, using genetic variables to compare different protocols and subjects.

## Figures and Tables

**Figure 1 behavsci-14-01084-f001:**
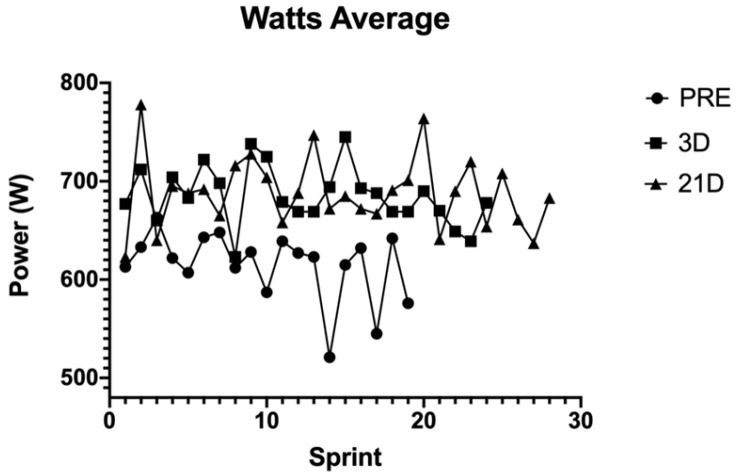
Power output of every sprint in the different tests.

**Figure 2 behavsci-14-01084-f002:**
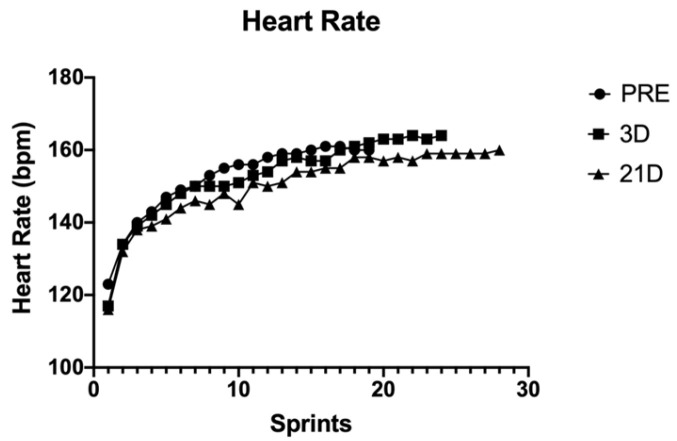
Heart rate (beats per minute) during every sprint in the different tests.

**Figure 3 behavsci-14-01084-f003:**
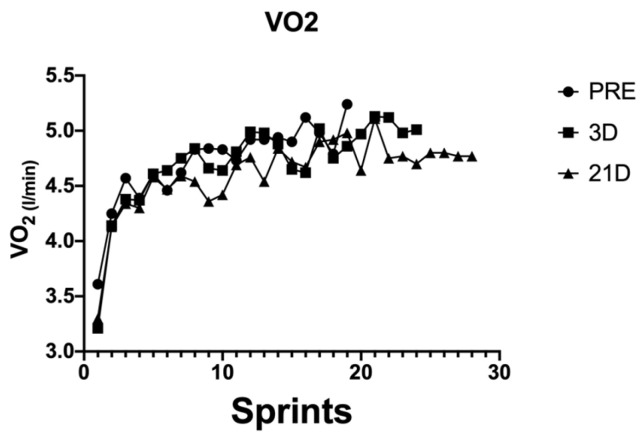
Volume of oxygen consumption (l/min) during every sprint in the different tests.

**Figure 4 behavsci-14-01084-f004:**
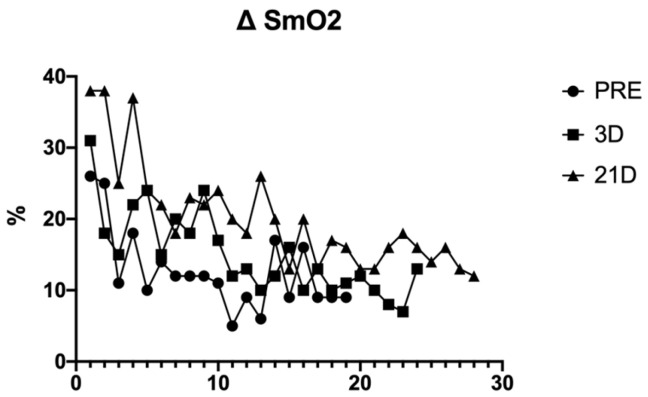
Difference between maximum and minimum SmO_2_ (%) during every sprint in the different tests.

**Table 1 behavsci-14-01084-t001:** Protocol.

Week 1	Mon	Tue	Wed	Thur	Fri	Sat	Sun
LHLi	HYP	HYP	HYP	HYP	HYP	NOR	NOR
IHT	2				2		
IHE							
Week 2	Mon	Tue	Wed	Thur	Fri	Sat	Sun
LHLi	HYP	HYP	HYP	HYP	HYP	NOR	NOR
IHT		2				3	
IHE							
Week 3	Mon	Tue	Wed	Thur	Fri	Sat	Sun
LHLi	HYP	HYP	HYP	HYP	HYP	NOR	NOR
IHT			3				3
IHE							
Week 4	Mon	Tue	Wed	Thur	Fri	Sat	Sun
LHLi	HYP	HYP	HYP	HYP	HYP	NOR	NOR
IHT			2				2
IHE							

LHLi > HYP = days when the participant was exposed to ambient hypoxic environment. NOR = rest days from hypoxic exposure. IHT -> This number expressed the blocks of sprints that the athlete performed. IHE -> Black background color shows the days where this protocol was used.

**Table 2 behavsci-14-01084-t002:** Variables for analysed performance in RSA test: number of sprints (N_Sprint_), 10” average of power output (W_Avg_), 10” average of muscular oxygen saturation (Avg ΔS_m_O_2_), Fatigue Index (FI) and percentage decrement score. (S_Dec_).

	N_Sprint_	W_Avg_	Avg ΔS_m_O^2^	FI	S_Dec_
PRE	19	615 ± 35 W	12.63 ± 5.6%	21.4%	8.7%
POST-D3	24	685 ± 29 W	15.04 ± 5.8%	16.4%	8.0%
POST-D21	28	683 ± 49 W	20.17 ± 7.4%	34.6%	12.2%

## Data Availability

The original contributions presented in the study are included in the article/Appendix A, further inquiries can be directed to the corresponding authors.

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
