# Peer review of "Effects of a Combined Method of Normobaric Hypoxia on the Repeated Sprint Ability Performance of a Nine-Time World Champion Triathlete: A Case Report"

_behavsci, 2024, doi:10.3390/bs14111084_

Round 1

Reviewer 1 Report (New Reviewer)

Comments and Suggestions for Authors

1.      Line 10: please improve the phrasing, maybe: Elite athletes are an underrepresented population in scientific studies. Overall, the paper needs to improve phrasing and grammar.

2.      Line 29: There are papers that study elite athletes. Your statement is vague and needs rewriting to focus on the specific.

3.      The lack of a control group is acknowledged but it is a major limitation. This weakens the study's ability to attribute observed changes/ improvements solely to the hypoxia protocol. Please discuss potential ways to mitigate this limitation in future studies, such as using historical control data from the literature if any.

4.      The discussion (first paragraph) is very speculative where various mechanisms behind the observed improvements in RSA performance were discussed. But they do not back these claims with sufficient references or in-depth explanation. More citations and evidence should be provided to support these speculations.

5.      Line 215-219: The conclusion needs to be rewritten for clarity. Please focus more on the broader implications.

Comments on the Quality of English Language

Please refer the previous section.

Author Response

Dear reviewer,

Thank you very much for all your recomendations to improve the quality of the work. I will explain all the points that you have reviewed with all the changes that I have made in the paper:

1. Line 10: please improve the phrasing, maybe: Elite athletes are an underrepresented population in scientific studies. Overall, the paper needs to improve phrasing and grammar.

We have changed the phrasing as you told med. Line 10

2. Line 29: There are papers that study elite athletes. Your statement is vague and needs rewriting to focus on the specific.

We have changed the phrase between Line 30 and 31. There are papers that study elite athletes but not enough scientific literature with the characteristic of the athlete we have measured.

"Some scientific studies have analysed elite triathletes, but there is insufficient scientific literature on top-class elite triathletes."

3. The lack of a control group is acknowledged but it is a major limitation. This weakens the study's ability to attribute observed changes/ improvements solely to the hypoxia protocol. Please discuss potential ways to mitigate this limitation in future studies, such as using historical control data from the literature if any.

One of the main limitation of this study is the lack of control group, but It is difficult to compare another athlete with this characteristics but we have included more references of different studies which have compared hypoxia vs normoxia condition. Reference 8

4. The discussion (first paragraph) is very speculative where various mechanisms behind the observed improvements in RSA performance were discussed. But they do not back these claims with sufficient references or in-depth explanation. More citations and evidence should be provided to support these speculations.

We have included more references about that (5,8,12,13) to compare the result of this case report with other studies. 

5. Line 215-219: The conclusion needs to be rewritten for clarity. Please focus more on the broader implications.

I have change the conclusion and include more references to compare the results with similar cases.

We have sent the pape to a profesional proofreading service to improve the quality of the english as you suggest.

Thank you very much for all your contributions to the work.

Reviewer 2 Report (New Reviewer)

Comments and Suggestions for Authors

While this case study encompasses the physiological and performance outcomes of an elite triathlete following a part of his repeated sprint in hypoxia training, major concerns should be addressed before the manuscript is allowed into further stages of the publication process. Even though the authors should be congratulated for publishing data obtained from the population group of elite athletes, the usage of English must be improved. The mechanisms underpinning the increase in performance following the protocol are not discussed to a sufficient extent. The authors have not commended on the timing of competition following the RSH protocol despite their two (out) of three measurement timepoints being realized after the training protocol. Referencing is scarce; plenty of points raised should be defended by reference to published literature.

Comments on the Quality of English Language

Not of an appropriate level.

Author Response

Thank you very much for all your considerations to our work. We have made some changes in the conclusion part with more references to clarify some parts and use the comparison of other studies between hypoxia and normoxia protocol. One of the main problems of this works is to compare the subject but the specific characteristic makes it difficult. We are focus on physiological variables to confirm the change in the performance because the result of a competition is quite subjective, depends on the specific characteristic of the course, ambiental conditions, other athletes etc. but the athlete who was measured after the protocol get a World Triathlon Championship Series and Ironman 70.3 World Champhionship win. We have send the paper to a professional proof reading service to improve the english quality as you suggest.

Thank you very much for all your considerations and the contributions to improve the quality of the work

Round 2

Reviewer 1 Report (New Reviewer)

Comments and Suggestions for Authors

The authors have sufficiently addressed all of my previous comments. 

This manuscript is a resubmission of an earlier submission. The following is a list of the peer review reports and author responses from that submission.

Round 1

Reviewer 1 Report

Comments and Suggestions for Authors

The titled study Effects of a combined methods in normobaric hypoxia in RSA performance of nine-time world champion triathlete is interesting but I have some considerations:

INTRODUCTION

The introduction needs to be improved. It does not explain the relationship of RSA on the performance of a triathlete. That is, the RSA concept comes from team sports and refers to the capacity/ability to repeat sprints or maximum actions intermittently and with incomplete recovery, something completely different from what happens in triathlon, which is a cyclical sport. This should be explained to understand the justification for the intervention.

METHODS AND RESULTS

The methodology and results are well described. However, there are several limitations as mentioned by the authors. The main one is that it cannot be demonstrated that the improvements are due to the use of hypoxia training, therefore it must be corrected and talk about combined training or an implement. That is to say, an athlete will never stop training to introduce other techniques, but rather will incorporate them and use them as an implement with the aim of seeking an extra improvement in performance. This is the case of hypoxia training. In fact, these results could not be generalized since the effect on another triathlete with a different training level stimulus could have another result.

DISCUSION

This should be taken up in the discussion. The relationship between RSA and triathlon remains unknown, even more so when the athlete who is the subject of the study has not competed in (or at least has not won) the sprint category. The discussion should be improved by looking at how this improvement in sprints has a transfer to triathlon, since the most normal thing would have been to assess performance with a test that better reflects triathlon.

Author Response

Thank you very much for all your considerations. In this document I add the notes that you are considering and I attached to this the document with some changes in relation to your considerations.

Introduction

The relationship between RSA and contemporary triathlon is cause the changes in the bike courses. Now the bike courses are more technical and hilly which are challenging the RSA performance of the athlete. The capacity/ability to repeat a maximum sprint in every lap of the competition in every corner of the competition converts a triathlon bike course in a RSA situation. 

"The inclusion of hills or more ‘technical’ bike courses may change the physiological demands during draft-legal competitions. Smith et al.[40] examined the power output during the cycle stage of an ITU triathlon event using the SRM crank system enabling work output to be quantified. They concluded that power output fluctuated markedly during the cycle stage with athletes generating between zero to well in excess of the maximal levels."

David J. Bentley, Grégoire P. Millet, Verónica E. Vleck… (2002). Specific Aspects of Contemporary Triathlon. , 32(6), 345–359. doi:10.2165/00007256-200232060-00001 

Methods and result

The results of this study could not be generalized because it is a case report study but It confirms that this combining methods protocol works on an athlete with this characteristics and could promote a new investigation topic to works on a bigger group.

Discussion

In World Triathlon Ranking there are two differents distance with drafting legal (sprint distance and olympic distance) which has common characteristic during the bike course. The bike courses are more technical and hilly which are challenging the RSA performance of the athlete to stay in the group and save energy with drafting. (Line 161-163) in the manuscript

The improve in the competition performance was significant cause the athlete won the next World Triathlon Championship Series just after the protocol.

Reviewer 2 Report

Comments and Suggestions for Authors

This manuscript provides a useful information from an elite triathlete. However, some contents in the section of Materials and Methods are not clear and complete enough. It is recommended that the author to clarify and correct them.

1. Study design describes “The triathlete had performed a total of 3 weeks of a combined protocol of normo-baric hypoxia.” Variables were measured before the protocol (PRE), just after (3D) and 21 days after the end of the protocol (21D). However, Table 1 shows a total of four weeks (W1-W4) (28 days) of training. I don’t understand why the post-test was on day-21.

2.     Table 1also shows the subject received LHTLi 5 days per week (the text express 4 days in line 58 ?), 8 times within 4 weeks, and IHE during the whole reporting period. Therefore, the subject in this case report may receive 3 conditions of LHTLi, IHE, and IHT at the same time. I suggest using the term of “protocol” instead of “method” for the 3 interventions. And please describes the times of each protocol into the study design.

3.      Table 2 shows FI and SDec in 21D higher than 3D and PRE. The discussion describe “this study confirms that the hypoxic stimulus increases performance in the fatigue condition (lines 170-171)”. However, the numbers of sprint in PRE, 3D, and 21D is 19, 24, and 28. In my opinion, the higher fatigue is due to more sprint. I don't agree with this statement of “increases performance in the fatigue condition”.

4.      Please provide a clearer abbreviation in the horizontal column of Table 1, such as Week-1, Mon., Tue., ….Sun..

5.      I wonder the data of FI is × 100% ? And the same as SDec?

6.      Please add the word “6 sites” into line 52 for the description of “a total of skinfolds measurements of 45.5 mm”.

7.      Abstract express the data as PRE 19 vs 3D 24 vs 21D 28, it can easily be confused. I suggest modifying to “PRE vs 3D vs 21D: 19 vs 24 vs 28).

8.      Since this manuscript is a case study of an elite athlete, It is recommended that the conclusion modified to “This work “may” useful to improve power output…” in the last sentence of the abstract.

Comments on the Quality of English Language

It's unnecessary to capitalize the first letter of nouns with abbreviations in the text. And minor editing of English language required.

Author Response

Thank you very much for all your considerations. In this document I add the notes that you are considering and I attached to this the document with some changes in relation to your considerations.

  1. Study design describes “The triathlete had performed a total of 3 weeks of a combined protocol of normo-baric hypoxia.” Variables were measured before the protocol (PRE), just after (3D) and 21 days after the end of the protocol (21D). However, Table 1 shows a total of four weeks (W1-W4) (28 days) of training. I don’t understand why the post-test was on day-21.

There is a mistake in the text, The triathlete performed a total of 4 weeks, now I solved it. The reason of the post test after 21 days of the training protocol is because there is a new phase of adaptation between 15 and 21 days.

“On return to sea level after an altitude training camp, three phases have been observed by coaches (figure 2). So far, however, these are not fully supported by the scientific evidence and are therefore under debate:

  • A positive phase observed during the first 2–4 days, but not in all athletes.
  • A phase of progressive reestablishment of sealevel training volume and intensity. The probability of good performance is reduced.
  • 15–21 days after return to sea level, a third phase is characterized by a plateau in fitness. The optimal delay for competition is during this third phase, although some athletes reach their peak performance during the first phase.”

Millet GP, Roels B, Schmitt L, Woorons X, Richalet JP. Combining hypoxic methods for peak performance. Sports Med. 2010 Jan 1;40(1):1-25. doi: 10.2165/11317920-000000000-00000. PMID: 20020784.

  1. Table 1also shows the subject received LHTLi 5 days per week (the text express 4 days in line 58?), 8 times within 4 weeks, and IHE during the whole reporting period.

The text is incorrect, I have changed it. The athlete sleeps 5 days per week in hypoxic condition and rest 2 days in normoxia. The IHE protocol is exactly how you say it, the athlete performs the IHE protocol every day.

  1. Therefore, the subject in this case report may receive 3 conditions of LHTLi, IHE, and IHT at the same time. I suggest using the term of “protocol” instead of “method” for the 3 interventions. And please describes the times of each protocol into the study design.

I paste the part of the paper where I write all the questions that you asked me.

The study started at the beginning of the triathlete season. During the protocol, the triathlete developed an average training of 25,000 m of swim, 400 km of bike and 90 km of run. The triathlete had performed a total of 4 weeks of a combined protocol of normobaric hypoxia. The first method was Live High – Train Low interspersed (LHTLi). The participant slept 5 days in a row with a SpO2, which increased progressively to up to between 94% and 90% of SpO2, then he rested for 2 days of sleep in hypoxic conditions. The second protocol was intermittent hypoxic exposure (IHE), which consisted of giving a stimulus of 14.40 to 13.20 SpO2 every day, decreasing progressively during the protocol to leave the SpO2 of the triathlete between 88% and 90%. The last protocol was intermittent hypoxic training (IHT) with a total of eight sessions which was conformed by 2–3 sets of five sprints all out of 10” with a rest of 20” between sprints and between sets of 5’ (first three sessions were 2 blocks, then three sessions of 3 blocks and finally 2 sessions of 2 blocks). Between sessions the triathlete rested 72 hours of RSH. The SpO2 during the training was between 13.5% and 14.0%. The protocol was performed totally and without any changes without any adverse or unanticipated event (Table 1).

  1. Table 2 shows FI and SDecin 21D higher than 3D and PRE. The discussion describe “this study confirms that the hypoxic stimulus increases performance in the fatigue condition (lines 170-171)”. However, the numbers of sprint in PRE, 3D, and 21D is 19, 24, and 28. In my opinion, the higher fatigue is due to more sprint. I don't agree with this statement of “increases performance in the fatigue condition”.

The fatigue index only compares the best and the worse sprint, so the number of sprints does not affect to the FI. The sentence that you have cited is speaking about the performances of the last sprints in the test. (Figure 2)

The power output decreased slightly 21 days after the protocol had finished (685 vs 683) but the triathlete performed four more sprints, and after sprint number 18 the power output was bigger at 21D than at 3D training (3D = 666 vs 21D = 686) so this study confirms that the hypoxic stimulus increases performance in the fatigue condition.

  1. Please provide a clearer abbreviation in the horizontal column of Table 1, such as Week-1, Mon., Tue., ….Sun..

Solved

  1. I wonder the data of FI is × 100% ? And the same as SDec?

Yes both of them are percentage.

  1. Please add the word “6 sites” into line 52 for the description of “a total of skinfolds measurements of 45.5 mm”.

Solved

  1. Abstract express the data as PRE 19 vs 3D 24 vs 21D 28, it can easily be confused. I suggest modifying to “PRE vs 3D vs 21D: 19 vs 24 vs 28).

Solved

  1. Since this manuscript is a case study of an elite athlete, It is recommended that the conclusion modified to “This work “may” useful to improve power output…” in the last sentence of the abstract.

Solved

Round 2

Reviewer 1 Report

Comments and Suggestions for Authors

Dear authors,

RSA is the ability to repeat sprints and is typical of team sports, so in my first report I wanted to know how this occurs in a triathlete because the efforts are not intermittent. It may be the case that some rise appears but it would not be enough to say that they are intermittent efforts maintained over time. On the other hand, the fact that there is no comparison with another triathlete who has not worked with this hypoxia programming does not allow us to confirm that the results are due to it. Finally, the differences that appear in the test cannot be said to be significant (since statistics do not allow it to be confirmed) and in some cases it could be the effect of fatigue.

Author Response

Dear reviewer,

Thank you very much for your comments. It will help to improve the quality of the work. I have make some changes related to your suggestions.

RSA and relationship with triathlon

  • RSA is tipical of team sport and of course in triathlon is not exactly the same efforts but with the changes in the bike courses, the triathlon has changed a lot the bike efforts fluctuated markedly between zero to well in excess of the maximal levels.

David J. Bentley, Grégoire P. Millet, Verónica E. Vleck… (2002). Specific Aspects of Contemporary Triathlon. , 32(6), 345–359. doi:10.2165/00007256-200232060-00001

The study measure a rsa test to know how this protocolo affects to the ability to performs rsa efforts, of course in a real competition situation won't be as much as the test shows, but if the athlete improve this ability he will be prepare to resist more attacks, to attack or just to finish the bike sector in better conditions and has a better running performance.

Statistical analysis

The athlete is a world class athlete with singular characteristics and a really difficult competition calendar and specific training plan. The specific characteristics of the athlete makes imposible compare this data to the same conditions data. This study is the first combining hypoxia method to a world class triathlete so the objective was to give this data to the coaches to start doing the method with their athletes. Future studies can make a comparison to this one and can measure the diferences between season and athlete.

Thank you very much for your suggestions, they will improve a lot the quality of the article.

Reviewer 2 Report

Comments and Suggestions for Authors

1.      After the author's explanation, I now know that D21 is 21 days “after the end” of the protocol, I think it should be express as POST-D21, as well as use the term of POST-D3 to instead of D3.

2.      Since the last teste was conducted at 21 days after the workouts had finished, please describe the training/exercise/competition situation of the triathlete after return to sea level. And describes that 15-21 days after may reach peak performance with the citation of the reference of Millet et al. (2010) into the text.

3.      The formulations of FI and SDec should be ×𝟏𝟎𝟎%, and please adds unit of all variables in Table 2.

Comments on the Quality of English Language

Minor editing of English language required

Author Response

Thank you very much for all your considerations. I have changed all the suggestions that you make to the document, I have attached the new version of the document and I explain in this text where to find them.

1. After the author's explanation, I now know that D21 is 21 days “after the end” of the protocol, I think it should be express as POST-D21, as well as use the term of POST-D3 to instead of D3.

I have changed the way to call it in all the document (abstract, methods, results and discussion)

2. Since the last teste was conducted at 21 days after the workouts had finished, please describe the training/exercise/competition situation of the triathlete after return to sea level. And describes that 15-21 days after may reach peak performance with the citation of the reference of Millet et al. (2010) into the text.

I have include the training load in the study design (Line 63-66)

I have include the reference of Millet et al. (2010) in the assessment part (Line 89-90)

3. The formulations of FI and SDec should be ×???%, and please adds unit of all variables in Table 2.

I have include all the units in table 2 and change the way to call all the moments of the test.

Thank you very much for all your suggestions, It really helps to improve the work.
